# Proximity to Facilities and Its Association with the Health-Related Habits of Functionally Independent Older Adults

**DOI:** 10.3390/ijerph17228677

**Published:** 2020-11-23

**Authors:** Mónica Machón, Kalliopi Vrotsou, Isabel Larrañaga, Itziar Vergara

**Affiliations:** 1Grupo de Atención Primaria, Instituto de Investigación Sanitaria Biodonostia, 20014 San Sebastián, Spain; kalliopi.vrotsoukanari@osakidetza.eus (K.V.); itziar.vergaramitxeltorena@osakidetza.eus (I.V.); 2Red de Investigación en Servicios de Salud en Enfermedades Crónicas (REDISSEC), 48902 Baracaldo, Spain; 3Instituto de Investigación en Servicios de Salud Kronikgune, 48902 Baracaldo, Spain; 4Instituto de Investigación Sanitaria Biodonostia, 20014 San Sebastián, Spain; mlarranagapadilla@gmail.com; 5Departamento de Salud, Delegación Territorial de Gipuzkoa, Gobierno Vasco, 20010 San Sebastián, Spain

**Keywords:** older adults, neighborhood environment, physical activity, diet, self-perceived social life

## Abstract

The aim was to examine how proximity to facilities, as a component of community determinants, is associated with the health-related habits of functionally independent community-dwelling older adults. This was a cross-sectional study. Data were collected by face-to-face interviews. Participants were >65 years old, living in 15 municipalities of Gipuzkoa (Basque Country, Spain). Proximity to park-green spaces, cultural-sport centers, market-food stores, retirement associations, religious centers, primary care centers and hospitals was explored. Sociodemographic variables and health-related habits (diet, physical activity and self-perceived social life) were collected. Logistic regression models were performed. The sample comprised of 634 individuals (55% women; mean age: 74.8, SD 6.7 years). Older age (odds ratio-OR: 0.94, 95% CI: 0.91–0.97) was associated with lower physical activity, while being male (OR: 1.71, 95% CI: 1.08–2.68) and proximity to park-green spaces (OR: 1.64, 95% CI: 1.03–2.61) were related to more physical activity. Individuals with good self-perceived health (OR: 3.50, 95% CI: 1.82–6.74) and religious centers within walking distance (OR: 2.66, 95% CI: 1.40–5.04) had higher odds of a satisfactory social life. Encouraging the creation of park-green spaces and leisure centers near residential areas can assist in promoting physical activity and improving the social life of older adults.

## 1. Introduction

Aging is a complex process that can be affected by a myriad of factors, including the environment [1]. Previous evidence from international studies has found that environmental neighborhood conditions have an impact on the diet, physical activity and social participation of older people [2,3,4,5]. Overall, having community resources, such as grocery stores and cultural-sport centers, near residential areas is associated with an acquisition of healthier lifestyle habits [2,3,4,5]. In Spain, few studies have been conducted on this subject and, in general, they do not focus on older adults [6,7,8]. The findings from some countries may not be generalizable to others, as the built environment characteristics and the way they are used and experienced may vary depending on the cultural context [3]. Considering the meaning and identity of urban spaces is important because they contribute to one’s own identity, health, sense of community and sense of place [9]. Therefore, it is necessary to understand the level and form of attachment and meanings associated with places to unravel their effects on each community. More research on this very topic is needed in Spain. Furthermore, a better understanding of the potential effect that the environment may have on the health-related habits of older adults is a key element in identifying points of improvement. Such understanding can aid in designing more age friendly cities, defined as those that “encourage active ageing by optimizing opportunities for health, participation and security in order to enhance quality of life as people age” [10]. It can also be of great value in developing health promotion interventions based on the real needs of this population group.

The purpose of this study was to examine whether the proximity to facilities was associated with the health-related habits of functionally independent community-dwelling older adults.

## 2. Materials and Methods 

### 2.1. Study Population

This was a cross-sectional study of functionally independent people over 65 years of age living in the province of Gipuzkoa (Basque Country, Spain). Participants were included if they had a Barthel index ≥95 points [11,12] and did not present cognitive impairment according to the Short Portable Mental Status Questionnaire of Pfeiffer [13,14]. This test contained ten items with “right/wrong” type answers (e.g., what day is it today?, who is the current president?). Each “wrong” answer obtained one point. Normal cognitive status was considered ≤2 errors for participants who could write and read or ≤3 for illiterate individuals. A detailed description of the study population selection has been reported in previous works [15,16]. Briefly, participants from 15 municipalities with different sizes (i.e., <2000, 2001–10,000, 10,001–50,000 and >50,000 people) were included in the study. The sample was representative of the province as far as age and sex were concerned. Household interviews were carried out by trained interviewers. The data were collected from January 2013 to February 2013.

### 2.2. Variables

Sociodemographic data (age, sex and level of education) were recorded. Self-perceived health was assessed with an item, “Overall, you would say that your health is…”, and its five response options were grouped into two categories: good (excellent, very good, good) and poor (fair, poor) [17]. The health related habits examined were: diet, physical activity and self-perceived social life. Frequency of fruit, vegetable, milk-cheese-yogurt and fish intake was asked. The participants were considered as having an adequate diet if they ate fruit, vegetables and milk-cheese-yogurt ≥2 times per day and fish ≥2 times per week [18,19]. Physical activity was assessed by asking if they had performed any physical activity during the last two weeks. Social life was self-evaluated with a single item and responses were categorized as satisfactory (very satisfactory, satisfactory) versus unsatisfactory (unsatisfactory, very unsatisfactory) [17].

Neighborhood environment is a complex construct and different elements can be measured as part of it [1]. In this study, the proximity to a number of facilities was assessed: park-green space, cultural-sport center, market-food store, retirement association, religious center, primary care center and hospital. For each one of these, the participants had to respond to the question: “From your habitual home, how do you usually get to the following places?” [20]. The six answer options (walking less than 15 min; walking more than 15 min; by public transport; by taxi or car; I never go because it does not exist; I never go) were re-coded as facility within walking distance: yes, no. 

### 2.3. Statistical Analysis

Continuous variables are presented with means and standard deviations (SD). Categorical variables are described with frequencies and percentages (%). The Student’s t-test and Chi-square or Fisher’s exact tests were used for the comparison of continuous and categorical variables, respectively. Univariate and multivariate binary logistic regression models were fitted. The variables of age and sex were entered in all multivariate models. Variables with a *p*-value < 0.10 in the univariate stage were included in the multivariate analysis phase. Both backward and forward regression models were performed. The multivariate model estimations are presented with odds ratios (OR) and 95% confidence intervals (95% CI) along with their corresponding *p*-values. *p*-values < 0.05 were considered to be statistically significant. The Hosmer–Lemeshow test results, the area under the curve (AUC) and R-square values are provided for the final models. Statistical analyses were carried out with the SAS 9.3 software (SAS Institute, Cary, NC, USA).

## 3. Results

The sample was composed of 634 individuals (55% women), with a mean age of 74.8 (SD 6.7) years. Eighty-two percent had primary or lower educational level. Only 15% of the sample had an adequate diet, whereas the majority performed physical activity in the last two weeks (84%) and reported a satisfactory social life (93%) (Table 1). Regarding the proximity to facilities, most people accessed park-green space (75%), market-food store (81%), religious center (67%) and primary care center (83%) by walking. 

Diet, physical activity and self-perceived social life were associated with sociodemographic characteristics, self-perceived health and proximity to facilities at the univariate analysis. The level of education was the only variable associated with diet at this stage (*p* < 0.010), but this effect was not maintained at the multivariate model.

Thus, multivariate models are presented for physical activity and self-perceived social life (Table 2). Considering physical activity, older age was associated with a lack of physical activity during the last two weeks (OR: 0.94, 95% CI: 0.91–0.97). On the other hand, being male and having a park-green space within walking distance were related to greater odds of performing physical activity (OR: 1.71, 95% CI: 1.08–2.68 and OR: 1.64, 95% CI: 1.03–2.61, respectively). As far as self-perceived social life was concerned, individuals who reported good health (OR: 3.50, 95% CI: 1.82–6.74) as well as those who went walking to the religious center (OR: 2.66, 95% CI: 1.40–5.04) were more likely to have a satisfactory self-perceived social life.

## 4. Discussion

In the present study, based on a sample of functionally independent older adults, having community resources within walking distance was associated with performing physical activity and enjoying a satisfactory social life. A certain amount of association was found between these two outcomes (*p* = 0.031). However, the fact that different predictive variables eventually emerged in the respective models highlights the individual value of each outcome in the process of assessing proximity to facilities. Comparing the current findings with previous works is a difficult task, mainly due to the differences in the definition and measurement of variables, characteristics of the study population or statistical analysis. In relation to physical activity, our results are in agreement with those of Hanibuchi et al. [3], which reported a positive association between population density and the presence of parks or green spaces and the frequency of sports activity of Japanese older adults. Moreover, in a systematic review and meta-analysis, access to park/public open spaces was positively associated with physical activity of older adults [2]. Considering social life, similar to our study, a published review indicated a positive association between social participation and proximity to resources and recreational facilities [4]. In the present work, no association was found between diet and proximity to food market facilities. The vast majority of the sample (i.e., 81%) reported good access to the latter, meaning that proximity to these facilities is mainly assured for the local population. Previous evidence showed mixed results. In Nakamura et al. [5], low access to food stores was related to infrequent intake of fruits, vegetables, meat and fish in individuals aged ≥65 years. In the systematic review of Rahmanian et al. [21], 88% of the included studies (*n* = 24) found a statistically significant association between diet and built environment in adults >18 years old, but inconsistencies across the studies were also described. 

One of the main limitations of the study is its cross-sectional design, and the difficulty in establishing causal relationships. Furthermore, the data collection was carried out by 20 interviewers. They had experience in these kinds of interviews and they received additional training before initiating the fieldwork, which should have minimized the possible variability during data collection. Moreover, dietary habits were examined with vegetables, fruits, milk-cheese-yogurt and fish consumption. Other relevant food groups, such as white meat, legumes, eggs or cereals, were not included. However, in Rahmanian et al. [21], the majority of studies only considered fruit and vegetable consumption. Finally, the residential location is a decision that may have been related to certain variables used in the models. Information on this variable has not been collected, meaning that its effect cannot be assessed in our data. In any case, given the mean age of our sample, this information would most likely reflect past and not present conditions.

It is also important to describe the strengths of this work. It includes a fairly high number of functionally independent older adults living in the community. In order to develop action plans seeking to avoid or delay the onset of dependence, it is essential to examine the clinical and environmental factors related to it in a phase prior to that of dependence. In this sense, the age friendly city initiative, recently launched by the WHO, highlights the importance of urban features (e.g., urban spaces and buildings, transportation or housing) that can affect the health of older people [10]. Data that would help us to understand the complex interactions between environmental elements and the health-related habits of older populations are needed. Identifying neighborhood environmental aspects that call for improvement is a key goal in achieving and maintaining an active and healthy lifestyle.

## 5. Conclusions

The proximity to facilities showed a positive association with physical activity and self-perceived social life in a sample of functionally independent older adults. Enhancing the presence of park-green spaces as well as leisure centers near residential areas can promote an active lifestyle and healthy aging in such populations.

## Figures and Tables

**Table 1 ijerph-17-08677-t001:** Characteristics of functionally independent older adults according to their diet, physical activity and self-perceived social life.

Variables	Diet		*p*-Value	Physical Activity in the Last Two Weeks		*p*-Value	Self-Perceived Social Life		*p*-Value
	Adequate (*n* = 98)	Inadequate (*n* = 536)		Yes (*n* = 531)	No (*n* = 103)		Satisfactory (*n* = 591)	Unsatisfactory (*n* = 43)	
Age, years, mean (SD)	75.2 (7.0)	74.7 (6.6)	0.475	74.3 (6.5)	77.3 (7.0)	<0.0001	74.8 (6.6)	74.7 (6.8)	0.935
Sex			0.370			0.007			0.169
Women	58 (59)	291 (54)		280 (53)	69 (67)		321 (54)	28 (65)	
Men	40 (41)	245 (46)		251 (47)	34 (33)		270 (46)	15 (35)	
Level of education ‡			0.030			0.899			0.286
Primary or lower	72 (74)	447 (83)		435 (82)	84 (82)		482 (82)	37 (88)	
Secondary or higher	25 (26)	89 (17)		95 (18)	19 (18)		109 (18)	5 (12)	
Self-perceived health			0.248			0.379			<0.0001
Poor	22 (22)	94 (18)		94 (18)	22 (21)		98 (17)	18 (42)	
Good	76 (78)	442 (82)		437 (82)	81 (79)		493 (83)	25 (58)	
Park-Green space within walking distance ‡	-	-	-			0.015			0.020
No				122 (23)	35 (34)		140 (24)	17 (40)	
Yes				408 (77)	67 (66)		449 (76)	26 (60)	
Cultural-Sport center within walking distance	-	-	-			0.039			0.078
No				319 (60)	73 (71)		360 (61)	32 (74)	
Yes				212 (40)	30 (29)		231 (39)	11 (26)	
Market-Food store within walking distance			0.115	-	-	-	-	-	-
No	24 (24)	95 (18)							
Yes	74 (76)	441 (82)							
Retirement association within walking distance ‡	-	-	-	-	-	-			0.032
No							340 (58)	32 (74)	
Yes							248 (42)	11 (26)	
Religious center within walking distance ‡	-	-	-	-	-	-			0.002
No							184 (31)	23 (53)	
Yes							404 (69)	20 (47)	

Numbers are *n* (%) unless otherwise stated; -: The variable was not measured; ‡: 1 to 3 missing values in these variables.

**Table 2 ijerph-17-08677-t002:** Factors associated with physical activity and self-perceived social life in functionally independent older adults.

Variables	Physical Activity in the Last Two Weeks		Self-Perceived Social Life	
	OR (95% CI)	*p*-Value	OR (95% CI)	*p*-Value
Age, years, mean (SD)	0.94 (0.91–0.97)	0.0003	1.00 (0.96–1.05)	0.769
Sex		0.021		0.140
Women	1.00		1.00	
Men	1.71 (1.08–2.68)		1.65 (0.84–3.22)	
Self-perceived health	-	-		0.0002
Poor			1.00	
Good			3.50 (1.82–6.74)	
Park-Green space within walking distance		0.039	-	-
No	1.00			
Yes	1.64 (1.03–2.61)			
Religious center within walking distance	-	-		
No			1.00	
Yes			2.66 (1.40–5.04)	0.003

-: The variable is not part of the multivariate model; Physical activity: probability modeled is “Yes”, estimates are based on *n* = 632 participants. Model diagnostics Area Under the Curve (AUC) = 0.661; Max-rescaled R-Square = 0.069; Hosmer–Lemeshow *p*-value = 0.441; Self-perceived social life: probability modeled is “Satisfactory social life”, estimates are based on *n* = 631 participants; AUC = 0.716; Max-rescaled R-Square = 0.096; Hosmer–Lemeshow *p*-value = 0.581.

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
