# Peer review of "Proximity to Facilities and Its Association with the Health-Related Habits of Functionally Independent Older Adults"

_ijerph, 2020, doi:10.3390/ijerph17228677_

Round 1
Reviewer 1 Report
The introduction lacks theoretical foundation and scientific evidence to justify the information and argumentation shown.
The wording of the inclusion criteria is imprecise. For example, what score on the SPMSQP would indicate "no cognitive deficit"? or "three mistakes reading", but how much text or how many words did they have to read?
The groupings of the sample made for the variables diet, physical activity and perceived social life generate very disproportionate comparative groups, such as 43 participants compared to 591 participants.
How robust can the conclusions drawn from this study be taking into account these disproportions of participants in the comparisons between groups?
Author Response
The introduction lacks theoretical foundation and scientific evidence to justify the information and argumentation shown.
Answer: We have now included more text and references to justify the information given in the introduction (lines 41-50). Nonetheless, we would like to clarify that many relevant references are included in this section (i.e. references 1-9) and also that due to the word limitations that the journal indicates for brief reports, this section cannot be too extensive.
The wording of the inclusion criteria is imprecise. For example, what score on the SPMSQP would indicate "no cognitive deficit"? or "three mistakes reading", but how much text or how many words did they have to read?
Answer: Thank you for the comment. Effectively, this point was not well explained. We have now modified the text to better describe the type of questions of the scale. We hope that the current explanation does not leave room for misunderstandings (lines 59-62).
The Short Portable Mental Status Questionnaire of Pfeiffer is composed of ten items (e.g. day today; day of the week; how old are you?... ). All answers are either “right” or “wrong”. Wrong answers receive 1 point. Normal cognitive status is considered when the person has ≤2 errors for those who could read and write or ≤3 for illiterate individuals.
The groupings of the sample made for the variables diet, physical activity and perceived social life generate very disproportionate comparative groups, such as 43 participants compared to 591 participants.
Answer: In relation to diet, the participants were considered as having an adequate diet if they ate fruit, vegetable and milk-cheese-yogurt ≥2 times per day and fish ≥2 times per week. To define if a person has an adequate diet we have taken into account the food recommendations of the healthy eating pyramid of the Spanish Society of Community Nutrition (SENC) (SENC web; Aranceta-Bartrina et al.).
Physical activity was assessed by asking if they had performed any physical activity during the last two weeks. The possible answers were only “yes” or “no”. In this case, it was not possible to group the given answers in any other way.
Regarding social life, it was self-evaluated with a single item and responses were categorized as satisfactory (very satisfactory& satisfactory) versus unsatisfactory (unsatisfactory& very unsatisfactory). We believe that this is the best way to group the answers.
We did not want to define a priori proportionate comparative groups for these variables, but to provide a picture of the health related habits of the studied sample. The results obtained showed that only a 15% of the sample had an adequate diet, whereas the majority performed physical activity in the last two weeks (84%) and reported a satisfactory social life (93%). These findings indicate that for example diet need to be improved in this sample.
References:
-Sociedad Española de Nutrición Comunitaria Guía de la alimentación saludable para atención primaria y colectivos ciudadados Available online: https://www.nutricioncomunitaria.org/es/noticia/guia-alimentacion-saludable-ap.
-Aranceta-Bartrina, J.; Partearroyo, T.; López-Sobaler, A.M.; Ortega, R.M.; Varela-Moreiras, G.; Serra-Majem, L.; Pérez-Rodrigo, C. Updating the food-based dietary guidelines for the Spanish population: The Spanish society of community nutrition (senc) proposal. Nutrients 2019, 11, doi:10.3390/nu11112675.
How robust can the conclusions drawn from this study be taking into account these disproportions of participants in the comparisons between groups?
Answer: The final presented results have been based on multivariate binary logistic regression models, adjusted for age and sex on a sample of >600 individuals. The “dis-proportionality” that the reviewer worries about is not an issue here. These models do provide robust estimations, as they have >10 events per variable (Austin et al.). In any case, the presented models are not predictive, they are exploratory, as the goal of our article was to explore and present the data in a comprehensive way that will help us in understanding better the effect that proximity to facilities have over health related habits.
References:
-Austin PC, Steyerberg EW. Events per variable (EPV) and the relative performance of different strategies for estimating the out-of-sample validity of logistic regression models. Stat Methods Med Res. 2017 Apr;26(2):796-808.

Reviewer 2 Report
This interesting study uses data collected and analyzed in other reports to explore the how proximity to green space/parks is associated with physical activity and satisfaction with social life.
From my perspective the study has a serious analytic flaw that could be addressed by a more comprehensive analysis. I believe there is a significant consensus in the urban planning and urban form/physical activity literature that persons and households make residential location decisions based in part on proximity to amenities. The regression approach adopted in this article does not account for this. A more appropriate analysis would be a two-stage regression, the use of what economists call the Heckman procedure to address biases in the sample. In this case, a first stage model would include all but one of the variables considered to predict proximity to green space and other amenities of interest, and then in a second stage, the predicted proximity and all other potential covariates are considered as predictors of the dependent variable. Application of this approach might require the creation of a summary index of proximity.
One other concern, I think there have been a number of studies showing significant correlations between physical activity and social participation. The authors might also want to consider how their two dependent variables are linked and how they fit in the causal structure.
Author Response
Comments and Suggestions for Authors
This interesting study uses data collected and analyzed in other reports to explore the how proximity to green space/parks is associated with physical activity and satisfaction with social life.
From my perspective, the study has a serious analytic flaw that could be addressed by a more comprehensive analysis. I believe there is a significant consensus in the urban planning and urban form/physical activity literature that persons and households make residential location decisions based in part on proximity to amenities. The regression approach adopted in this article does not account for this. A more appropriate analysis would be a two-stage regression, the use of what economists call the Heckman procedure to address biases in the sample. In this case, a first stage model would include all but one of the variables considered to predict proximity to green space and other amenities of interest, and then in a second stage, the predicted proximity and all other potential covariates are considered as predictors of the dependent variable. Application of this approach might require the creation of a summary index of proximity.
Answer: Thank you for this comment. We do agree that people are likely to choose their place of residence based on the factors you mention; we have never claimed differently. Of course, location may also be affected by additional factors, like household economic situation. In this study, given that our sample was comprised of people >65 years of age, current location may have been affected by several factors many years ago, when most of these people probably bought their residence. All these, can only be discussed on theoretical grounds, given that no information related to the location decision has been collected as part of our study. In order to account for a phenomenon, this should somehow be measured or quantified. But, the location decision information is something we have no info on.
With the chosen analysis we are trying to see whether proximity to facilities affect diet, physical activity and self-perceived social life. The obtained results are reasonable, make sense and are supported by previously published literature. The logistic regression models we have applied in this case offer robust estimations and are a popular choice in similar studies (Wong et al.; Wu et al.; Dujardin et al.; Ribeiro-Ferreira F et al.; Chiang et al.; Hale L. et al.). The Heckman procedure that the reviewer suggests is usually used in econometric studies, for non-random samples. The authors of this work went into a great deal of effort to select a sample that was representative of the local population, and this is not an econometric study. Also, it is not clear to us, what would creating a summary index of proximity add to our results. It seems that the reviewer suggests an alternative way of analyzing the data, but we are sorry to say that do not see the point in applying such changes here. For all the above reasons we have opted for maintaining the analyses/results unchanged in this manuscript.
References:
-Wong M, Yu R, Woo J. Effects of Perceived Neighbourhood Environments on Self-Rated Health among Community-Dwelling Older Chinese. Int J Environ Res Public Health. 2017 Jun 7;14(6):614.
-Wu ZJ, Song Y, Wang HL, Zhang F, Li FH, Wang ZY. Influence of the built environment of Nanjing's Urban Community on the leisure physical activity of the elderly: an empirical study. BMC Public Health. 2019 Nov 6;19(1):1459.
-Dujardin C, Lorant V, Thomas I. Self-assessed health of elderly people in Brussels: does the built environment matter? Health Place. 2014 May;27:59-67.
-Ferreira FR, César CC, Andrade FB, Souza Junior PRB, Lima-Costa MF, Proietti FA. Aspects of social participation and neighborhood perception: ELSI-Brazil. Rev Saude Publica. 2018 Oct 25;52 Suppl 2(Suppl 2):18s.
-Chiang CC, Chiou ST, Liao YM, Liou YM. The perceived neighborhood environment is associated with health-enhancing physical activity among adults: a cross-sectional survey of 13 townships in Taiwan. BMC Public Health. 2019 May 7;19(1):524.
-Hale L, Hill TD, Friedman E, Nieto FJ, Galvao LW, Engelman CD, Malecki KM, Peppard PE. Perceived neighborhood quality, sleep quality, and health status: evidence from the Survey of the Health of Wisconsin. Soc Sci Med. 2013 Feb;79:16-22.
One other concern, I think there have been a number of studies showing significant correlations between physical activity and social participation. The authors might also want to consider how their two dependent variables are linked and how they fit in the causal structure.
Answer: In this work, we have studied different health related habits, and a certain amount of correlation between them is probably present. But, our focus was not the link between the outcomes. This would be a totally different study, not the one we have planned here. Our focus was exactly what we present: we wished to explore each outcome on its own, in order to better understand which proximity to facilities may affect each. Our results show the full picture in every case, while the age and sex adjusted models offer information of great value. Based on our observations proximity to facilities showed a positive influence on physical activity and self-perceived social life in a sample of functionally independent older adults. Enhancing the presence of park-green spaces as well as leisure centers near residential areas can promote an active and healthy aging in such populations.

Reviewer 3 Report
Would be good to add socio-economic status if possible, as that likely plays a role.
Also, might be helpful to add measured data (could people where a device to record steps) given restraints on self reported data.
Also, a focus group or survey may be helpful to add to reasoning to explain interesting findings. For example, why a religious place was a bigger predictor in physical activity than a park.
Author Response
Comments and Suggestions for Authors
Would be good to add socio-economic status if possible, as that likely plays a role.
Answer: Thank you for the comment. In the battery of questions, there was an item about monthly family income. As you can see in the table below, the percentage of missing was high, i.e. 23% of the sample. For this reason, we decided to use the educational level instead of the income as an indicator of socioeconomic position.
Income level is a sensitive information and people may be reluctant to provide it (Galobardes et al.). In addition, present income may not be an accurate measure of the total financial resources available to an older person (Robert et al.).
On the other hand, educational level is easy to obtain (Galobardes et al.).In our sample, no missing data were seen in this variable. Education can condition employment and income opportunities, and consequently the pension during retirement (Borrell et al.; Regidor et al.; Mejía-Lancheros et al.). Moreover, education tend to remain stable throughout life (Martínez-Sánchez et al.; Mejía-Lancheros et al.).
|
|
Total N =634 |
|
Monthly family income (€) |
|
|
≤1,500 |
345 (54%) |
|
≥1,500 |
145 (23%) |
|
Missing |
144 (23%) |
References:
-Galobardes B, Lynch J, Smith GD. Measuring socioeconomic position in health research. Br Med Bull. 2007;81-82:21-37.
-Robert S, House JS. SES differentials in health by age and alternative indicators of SES. J Aging Health. 1996 Aug;8(3):359-88.
-Borrell C, Marí-Dell’Olmo M, Rodríguez-Sanz M, et al. Socioeconomic position and excess mortality during the heat wave of 2003 in Barcelona. Eur JEpidemiol. 2006;21:633–40.
-Regidor E, Gutiérrez-Fisac J, Banegas J, et al. Association of adult socioeconomic position with hypertension in older people. J Epidemiol Community Health.2006;60:74–80.
-Mejía-Lancheros C, Estruch R, Martínez-González MA, et al. Impact of psychosocial factors on cardiovascular morbimortality: a prospective cohort study.BMC Cardiovasc Disord. 2014;14:135.
-Martínez-Sánchez E, Gutiérrez-Fisac JL, Gispert R, et al. Educational differences in health expectancy in Madrid and Barcelona. Health Policy. 2001;55:227–31.
Also, might be helpful to add measured data (could people where a device to record steps) given restraints on self reported data.
Answer: Thank for the suggestion. Unfortunately, in this study we do not have information about measured data, like a device to record steps. Budget limitations did not allow the inclusion of such measures.
Also, a focus group or survey may be helpful to add to reasoning to explain interesting findings. For example, why a religious place was a bigger predictor in physical activity than a park.
Answer: Thank you for the comment. This is definitely a suggestion that we will keep in mind for future studies. Once again, due to financial limitations and time restriction the project could not explore additional aspects of great relevance. We would also like to clarify that having a religious center within walking distance was associated with self-received social life, not physical activity. Having a park within walking distance was associated with higher physical activity.

Round 2
Reviewer 1 Report
Questions and doubts have been correctly ansewred.
Author Response
The reviewer said "Questions and doubts have been correctly ansewred".
Therefore, no new changes need to be done in the manuscript.
Reviewer 2 Report
The authors have rejected my concerns about the study.
As noted, residential location represents the results of a selection process in part related to all the other predictors of the two outcomes thus the resulting comparisons of sample groups by location is biased in unknown ways. The authors have not addressed this. The two dependent variables are clearly correlated. Showing results for each without exploring these relationships provides an incomplete analysis and potentially misleading policy judgements....The author(s) could have rejected any further analysis and provided some commentary on the issues raised by the review, but decided to just reject the recommendations.
Author Response
The authors have rejected my concerns about the study.
As noted, residential location represents the results of a selection process in part related to all the other predictors of the two outcomes thus the resulting comparisons of sample groups by location is biased in unknown ways. The authors have not addressed this. The two dependent variables are clearly correlated. Showing results for each without exploring these relationships provides an incomplete analysis and potentially misleading policy judgements....The author(s) could have rejected any further analysis and provided some commentary on the issues raised by the review, but decided to just reject the recommendations.
Answer: We are sorry to hear that the reviewer considered their concerns as “rejected”. This was not our intention. We went through all reviewer´s comments and replied in a very detailed way to each and every one of them. We thus argued all our decisions.
Nonetheless, in the current version of the manuscript we have now added a limitation addressing the residential location issue, as the reviewer suggests (lines 154-157).
As far as physical activity and self-perceived social life are concerned: they do present a certain amount of association (p=0.031). From our point of view this is not surprising at all, and it does not change neither the validity of the obtained results, nor the conclusions of this work. Again, we wish to clarify that the focus of this work has never been the association between the outcomes. However, following the reviewers suggestion we have now added more text in the discussion (lines 128-131).
We have tried to incorporate both reviewer concerns into the current manuscript. We hope that the current manuscript meets the standards of publication in your journal, and we are looking forward to your final decision.
Best regards.
